# Cytokine and Chemokine mRNA Expressions after *Mycobacterium tuberculosis*-Specific Antigen Stimulation in Whole Blood from Hemodialysis Patients with Latent Tuberculosis Infection

**DOI:** 10.3390/diagnostics11040595

**Published:** 2021-03-26

**Authors:** Ji Young Park, Sung-Bae Park, Heechul Park, Jungho Kim, Ye Na Kim, Sunghyun Kim

**Affiliations:** 1Department of Internal Medicine, Park Clinic, Busan 49267, Korea; aiteite1@naver.com; 2Department of Clinical Laboratory Science, College of Health Sciences, Catholic University of Pusan, Busan 46252, Korea; tjdqo2013@gmail.com (S.-B.P.); wt2626@naver.com (H.P.); jutosa70@cup.ac.kr (J.K.); 3Clinical Trial Specialist Program for In Vitro Diagnostics, Brain Busan 21 Plus Program, Graduate School, Catholic University of Pusan, Busan 49267, Korea; 4Department of Internal Medicine, Kosin University Gospel Hospital, Busan 49267, Korea

**Keywords:** latent tuberculosis infection, cytokine, chemokine, mRNA expression, hemodialysis

## Abstract

There have been few reports on the kinetics of hemodialyzed (HD) patients’ immune responses in latent tuberculosis infection (LTBI). Therefore, in the present study, messenger ribonucleic acid (mRNA) expression levels of nine immune markers were analyzed to discriminate between HD patients with LTBI and healthy individuals. Nine cytokines and chemokines were screened through relative mRNA expression levels in whole blood samples after stimulation with *Mycobacterium tuberculosis* (MTB)-specific antigens from HD patients with LTBI (HD/LTBI), HD patients without LTBI, and healthy individuals, and results were compared with the QuantiFERON-TB Gold In-Tube (QFT-GIT) test. We confirmed that the C-C motif chemokine 11 (CCL11) mRNA expression level of the HD/LTBI group was significantly higher than the other two groups. Especially, the CCL11 mRNA expression level of the >0.7 IU/mL group in the QFT-GIT test was significantly higher than the <0.2 IU/mL group in the QFT-GIT test and the 0.2–0.7 IU/mL group in the QFT-GIT test (*p* = 0.0043). The present study reveals that the relative mRNA expression of CCL11 was statistically different in LTBI based on the current cut-off value (i.e., ≥0.35 IU/mL) and in the >0.7 IU/mL group. These results suggest that CCL11 mRNA expression might be an alternative biomarker for LTBI diagnosis in HD patients.

## 1. Introduction

There have been few reports on the accurate prevalence of tuberculosis (TB) in high-risk groups, such as the elderly, patients with diabetes or chronic kidney disease, etc. According to the Korean Society of Nephrology end-stage renal disease (ESRD) registry data in 2018, TB incidence was 0.5% in hemodialyzed (HD) patients [1]. Patients with ESRD receiving dialysis is a risk factor of active TB, which is 25.3 times more prevalent in this group than in individuals with normal renal function [2]. This relationship is likely associated with alterations in the immune system, such as conversion from latent TB infection (LTBI) to active TB due to various factors including uremia.

Interferon gamma (IFN-γ) release assays (IGRAs) are currently used to identify LTBI cases and are useful even in immunocompromised hosts; however, they should be interpreted carefully because of low sensitivity and specificity in a high-burden setting [3]. Moreover, 7.7% of IGRAs display discordant results in a duplicated test, and most are located within a range near the cut-off value [4,5]. Additionally, there is a high proportion of IGRA reversion in serial follow-up studies, and lower positive IGRA response is associated with reversion [6,7]. Thus, some investigators have suggested the use of an alternative borderline zone instead of the cut-off value in commercial IGRA tests to avoid over-diagnosing LTBI [8]. However, little is known about the impact of impaired cellular immunity on dialysis patients in terms of IGRA results.

In addition to plasma IFN-γ level after *Mycobacterium tuberculosis* (MTB)-specific antigen stimulation, several cytokines and chemokines have been investigated as potential biomarkers for MTB infection and disease. However, sparse research on cytokines and chemokines in LTBI or different forms of TB in immunocompromised patients such as ESRD patients receiving dialysis has been conducted. More research is required to confirm the development of active TB in immunocompromised patients.

In the present study, therefore, messenger ribonucleic acid (mRNA) expression levels of nine immune markers including IFN-γ, tumor necrosis factor-α (TNF-α), interleukin-2 receptor (IL-2R), C-X-C motif ligand 9 (CXCL9), CXCL11, IFN-γ-induced protein 10 (IP-10), C-C motif chemokine 11 (CCL11), granulocyte-macrophage-colony stimulating factor (GM-CSF), and TNF receptor (TNFR) were analyzed using a quantitative reverse transcription polymerase chain reaction (qRT-PCR) TaqMan probe assay with whole blood samples after MTB-specific antigen stimulation from HD patients and healthy individual groups to identify biomarker candidates in the discovery stage toward the development of a diagnostic test to distinguish HD patients with LTBI from healthy controls (HCs).

## 2. Materials and Methods

### 2.1. Study Design and Population

A total of 113 human whole blood samples were collected and discriminated. All enrolled individuals were more than 19 years old with ESRD and were undergoing more than three months hemodialysis treatment or were HCs. Those with human immunodeficiency virus (HIV) infection, liver cirrhosis of Child–Pugh class C, cancer or autoimmune disease and those who had received chemotherapy within the last three months and who had any history of active TB treatment were excluded. HD patients without repeated QuantiFERON TB Gold In-Tube (QFT-GIT) tests (Qiagen, Hilden, Germany) within six months were excluded in the examination of the reversion and conversion in the HD group. LTBI was diagnosed based on positive results of the QFT-GIT test, with no symptoms or signs of active TB and no history of previous TB. HCs all displayed no symptoms or signs of active TB and no signs of active TB on chest radiographs. Whole blood bioresources of HD patients were collected from Kosin University Gospel Hospital Biobank, Busan, the Republic of Korea. All whole blood samples of HD patients were the remaining samples after the IGRA test, and the test was performed before hemodialysis.

### 2.2. Study Procedures

Demographic and clinical data were collected with an electronic medical chart review that included the following: sex, age, TB contact history, prior TB, Bacillus Calmette-Guérin (BCG) scar or vaccination history, co-morbidity diabetes mellitus (DM) and ischemic heart disease (IHD), smoking history, body mass index (BMI), chest X-ray (CXR), and LTBI treatment history if available. Prior TB contact was defined as a contact history of any kind with active lifelong TB patients. Prior TB was defined as a previous treatment history that included treatment for active TB. Findings suggestive of abnormal CXR were fibrotic infiltrates with pleural thickening or calcified nodules over the upper lung fields or other fibrotic lesions documented from previous TB.

Low and normal BMI were defined based on the categories used by the National Heart, Lung, and Blood Institute and the World Health Organization (WHO) (low BMI < 18.5 kg/m^2^ and normal BMI between 18.5 and 24.9 kg/m^2^).

### 2.3. QuantiFERON-TB Gold In-Tube (QFT-GIT) Assay

Peripheral whole blood samples were collected in three QFT-GIT collection tubes (nil, MTB-specific antigens (including ESAT-6, CFP-10, and TB7.7), and T-cell mitogen (PHA), which contain lithium heparin). Then, the QFT-GIT assay proceeded according to the manufacturer’s protocols.

An IFN-γ concentration of ≥0.35 IU/mL (after subtraction of nil control IFN-γ), following exposure to MTB-specific antigens, was considered positive for the QFT-GIT test; a concentration of <0.35 IU/mL was considered negative. If the IFN-γ response to T-cell mitogen was <0.5 IU/mL higher than that for the nil control, or >8 IU/mL higher than that for the nil control, the result was deemed indeterminate [9].

### 2.4. Total RNA Isolation and Reverse Transcription

In order to analyze relative mRNA expressions of IFN-γ, TNF-α, IL-2R, CXCL9, IP-10, CXCL11, CCL11, GM-CSF, and TNFR after MTB-specific antigen stimulation, blood cell pellets were treated using 500 μL of RNA/DNA stabilization reagent for Blood/Bone marrow (Roche Diagnostics, Mannheim, Germany). Then, MagNA Pure LC RNA Isolation Kit-High Performance kit (Roche Diagnostics) was used for RNA isolation according to the manufacturer’s protocols [10]. The concentration and purity of extracted total RNA were measured with a NanoDrop 2000 spectrophotometer (Thermo Fisher Scientific, Waltham, MA, USA), and the samples were stored at −80 °C until further use.

Next, complementary DNA (cDNA) was synthesized using a Moloney murine leukemia virus (M-MLV) reverse transcriptase kit (Invitrogen, Carlsbad, CA, USA) according to the manufacturer’s protocols. All reactions were performed using a SimpliAmp (Life Technologies, Carlsbad, CA, USA) thermal cycler.

### 2.5. qRT-PCR TaqMan Probe Assay Targeting Multiple Immune Marker mRNAs

Oligonucleotide primer sets and a TaqMan probe were designed using Primer3 software (Howard Hughes Medical Institute, MA, USA) for performing the qRT-PCR TaqMan probe assay targeting multiple immune markers including IFN-γ, TNF-α, IL-2R, CXCL9, IP-10, CXCL11, CCL11, GM-CSF, and TNFR. The oligonucleotide sequences of the PCR primer pairs and TaqMan probes were based on the National Center for Biotechnology Information (NCBI) reference mRNA or cDNA sequences of human target genes. Quantitative PCR was carried out with THUNDERBIRD^TM^ Probe qPCR Master Mix (TOYOBO, Osaka, Japan) using 3 μL of cDNA as the template in a total volume of 20 μL. The thermal cycling conditions were 10 min at 95 °C, followed by 40 cycles of 10 s at 95 °C and 30 s at 60 °C. All reactions were performed using an ABI 7500 FAST Real-time PCR system (Applied Biosystems, Waltham, MA, USA) and QuantSTUDIO^TM^ 7 Flex Real-time PCR system (Thermo Fisher Scientific). Relative quantification was performed by the 2-ddCT method; relative expression was calculated as the ratio between the mean threshold cycle (CT) values of the target genes and reference gene (GAPDH) in each stimulated sample in relation to a reference sample (not stimulated) [11].

### 2.6. Statistical Analysis

All data were expressed as median ± SD, and to further confirm statistical reliability, the interquartile range of the total data was used. Paired numeric data were compared using Student *t*-test or one-way analysis of variance (ANOVA) tests with Dunnett’s multiple comparisons. Chi-square analysis or Fisher’s exact test was used to evaluate differences in proportions. The kappa coefficient was calculated to check the correlation between two QFT-GITs. Statistical analyses were performed using SPSS version 24.0 (SPSS Inc., Chicago, IL, USA) and Graphpad Prism 8.2.1 software (Graphpad Software 8.2.1, San Diego, CA, USA). All tests of significance were two sided; *p* values <0.05 were considered statistically significant.

## 3. Results

### 3.1. Characteristics of the Study Population

The present study included 28 HD patients with LTBI (HD/LTBI), 47 HD patients without LTBI (HD/normal) and 48 HCs (see Table 1). The average age was 62.00 ± 11.18 years for the HD/LTBI group, 60.70 ± 12.46 years for the HD/normal group, and 23.94 ± 2.72 years for the HC group. HD groups were older than the HC group; however, there was no difference between HD/LTBI and HD/normal groups. The male proportion of the HD/LTBI group (18/28) was significantly higher than the HD/normal (16/47) and HC groups (21/48) (*p* = 0.038).

Previous TB contact and abnormal CXR lesion of the HD/LTBI group were more frequent than the other two groups (*p* = 0.002 and *p* = 0.005, respectively); however, BCG vaccination or scar of the HD/LTBI group was less frequent than the other two groups (*p* < 0.001). There was no significant difference in terms of underlying diseases such as DM and IHD between HD/LTBI and HD/normal groups. Low BMI in the HD/LTBI group was significantly more frequent than in the HD/normal group (*p* = 0.012).

The amount of IFN-γ released after T-cell mitogen stimulation in QFT-GIT was analyzed in HD and HC groups (Table 1). There was a statistically significant difference among the HD/LTBI, HD/normal, and HC groups (*p* = 0.005). In addition, the amounts of IFN-γ in the HD group were lower than in the HC group (*p* = 0.001).

### 3.2. QFT-GIT Test Results and Previous QFT-GIT Test Results within 6 Months

Most of the HD patients had repeated QFT-GIT tests within 4 to 6 months. QFT-GIT test responses according to positive (>0.7 IU/mL), borderline (0.2–0.7 IU/mL), and negative (<0.2 IU/mL) criteria were as follows: negative results were 97.9% in the HD/normal group, and positive results were 75% in the HD/LTBI group (see Table 2). Borderline results were 25% in the HD/LTBI group, 2.1% in the HD/normal group, and 4.2% in the HC group.

The previous QFT-GIT test response in HD patients was as follows: positive results were 92.9% in the HD/LTBI group and 21.3% in the HD/normal group. Negative results were 3.6% in the HD/LTBI group and 51.5% in the HD/normal group. Borderline results were 3.6% in the HD/LTBI group and 27.7% in the HD/normal group.

Based on the cut-off value of 0.35 IU/mL, the conversion rate was 3.1% (1/32) and the reversion rate was 37.2% (16/43). In addition, the reversion rate of the 0.35–0.70 IU/mL response group (85.7%, 6/7) was statistically higher than the >0.7 IU/mL response group (27.8%, 10/36) (*p* = 0.0162). According to the three QFT-GIT test response groups, 92.7% (13/14) of previous 0.35–0.70 IU/mL response cases and 25.0% (9/36) of previous >0.7 IU/mL response cases changed to negative results following the QFT-GIT test (see Figure 1, *p* < 0.001).

### 3.3. Comparison of Target Gene mRNA Expression Levels after Stimulation of MTB-Specific Ags for 24 h According to LTBI Diagnosis

There were no statistically significant differences for mRNA expression levels of IFN-γ, TNF-α, IL-2R, CXCL9, CXCL11, IP-10, GM-CSF, and TNFR (Figure 1a–i) among the three groups. However, there was a statistically significant difference in only CCL11 mRNA expression levels (*p* = 0.028): the mRNA expression level was higher in the HD/LTBI group (126.36 ± 406.25) than in the other two groups (HD/normal, 6.12 ± 17.45 vs. HC, 19.58 ± 55.86, Figure 1g). Within HD groups, the CCL11 of the HD/LTBI group was significantly higher than the HD/normal group (*p* = 0.0208).

### 3.4. Comparison of Target Gene mRNA Expression Levels after Stimulation of MTB-Specific Antigens for 24 h According to QFT-GIT Response Groups

There was no significant difference among the three QFT-GIT response groups in terms of IFN- γ, TNF-α, IL-2R, CXCL9, CXCL11, and IP-10 (Figure 2a–f). However, CCL11 mRNA expression levels in the > 0.7 IU/mL response group (168 ± 464) were significantly higher than in the <0.2 IU/mL response group (13.1 ± 42.5) and 0.2 to 0.7 IU/mL response group (2.55 ± 6.43, *p* = 0.0043, Figure 2g). Compared to CCL11 mRNA expression levels, GM-CSF mRNA expression levels in the 0.2 to 0.7 IU/mL response group (1,512,106 ± 4,781,492) were significantly higher than in the <0.2 IU/mL response group (626 ± 4129) and >0.7 IU/mL response group (15 ± 53.9, *p* = 0.0029, Figure 2h). TNFR mRNA expression levels in the <0.2 IU/mL response group (227,937 ± 720,687) were significantly higher than in the QFT-GIT-1 group (2.31 ± 4.44) and >0.7 IU/mL response group (4.01 ± 8.81, *p* = 0.0029, Figure 2i). Although there was no significant difference, the mRNA expression level of IL-2R in the >0.7 IU/mL response group was higher than in the less than 0.7 IU/mL groups (Figure 2c).

### 3.5. Comparison of Target Gene mRNA Expression Levels after Stimulation of MTB-Specific Antigens for 24 h According to QFT-GIT Response Groups in HD

There was a statistically significant difference in the mRNA expression of CCL11 among QFT-GIT test response groups in HD patients (*p* = 0.041), for which the expression levels in the >0.7 IU/mL response group were higher than those of the other two groups (Figure 3g).

The mRNA expressions of GM-CSF and TNFR differed significantly (*p* = 0.0132), and the expression levels of the 0.2 to 0.7 IU/mL response group were higher than the other two groups compared to CCL11 (Figure 3h,i). The median value of TNFR mRNA expression in the 0.2 to 0.7 IU/mL response group was higher than the other two groups; however, the median value of TNFR mRNA expression in the >0.7 IU/mL response group was higher than in the other two groups. Additionally, there were significant differences in GM-CSF and TNFR mRNA expression between the <0.2 IU/mL response group and 0.2 to 0.7 IU/mL response group (*p* = 0.0083). There was no significant difference among the three QFT-GIT response groups in terms of IFN- γ, TNF-α, IL-2R, CXCL9, CXCL11, and IP-10 (Figure 3a–f).

### 3.6. Comparison of Clinical Features and mRNA Expression of Cytokines and Chemokines between IGRA Reversion and Persistent Group in HD Group

There were no statistically significant differences between the reversion group (*n* = 16) and the persistent group (*n* = 58) in terms of sex, age, DM, IHD, and plasma IFN-γ protein level after T-cell mitogen stimulation and LTBI treatment. Additionally, there were no statistically significant differences in terms of cytokines and chemokines.

## 4. Discussion

Although an accurate detection of LTBI among HD patients is important, little is known about the kinetics of HD immune response in LTBI. Therefore, our study reveals several features concerning the differences in the MTB-specific antigen-stimulated mRNA expression of cytokines and chemokines between HD and HC groups.

The establishment of the TB granuloma is controlled by the synchronized expression of various chemokines. This synchronization has recently been shown to be critical in the control of TB disease along with CD4^+^ T-cell recruitment into the lung parenchyma [12,13,14].

At first, we checked the mRNA expression level of IFN-γ and TNF-α. According to one study, TNF-α and IFN-γ signaling have gained particular attention because their functions in host resistance to TB have been well documented in both mouse models and infected humans. Consistent with the previous study in patients with immune-mediated inflammatory disease, this study revealed that the amount of IFN-γ after mitogen stimulation was lower in the HD group than in the HC group [15]. Additionally, the IFN-γ of the mRNA expression level of the MTB-specific antigen-stimulated whole blood sample was not significantly different among the three groups according to LTBI diagnosis and QFT-GIT test response, although the mRNA expression levels of HC groups were higher than HD groups. However, for comparison within HD groups, the mRNA expression level of IFN-γ for the >0.7 IU/mL response group was higher than the <0.7 IU/mL response group. The IFN-γ mRNA expression decreased in HD patients, which may cause a false negative result for the QFT-GIT test, although there was no statistical significance.

Next, the IP-10 expression level was analyzed. IP-10 has been suggested as an alternative biomarker to IFN-γ in the QFT-plus format, as one study reported that IP-10 in response to TB1 was increased in subjects with LTBI compared to those with active TB [16]. However, this study discovered no significant difference between LTBI groups and normal groups, including HD and HC groups. IP-10 would not be useful in the diagnosis of LTBI, especially in immunocompromised patients.

In one study, CCL11 is produced by IFN-γ-stimulated endothelial cells and TNF-activated monocytes. It has been reported that CCL11, CCL24, and CCL26, which are produced by Th2 cells and other cells that induce Th2 development, are increased in TB patients compared to controls. It appears that TB suppresses Th1 and subsequently the classic function of macrophages by inducing chemokine expression [17]. Our study showed that the CCL11 mRNA expression level was significantly higher in the HD/LTBI group and >0.7 IU/mL response group, with no significance in terms of the receiver operating characteristic (ROC) analysis. Although CCL11 is little known in LTBI immune response, Yang et al. (2014) reported that the level of CCL11 continued to increase at the end of 12 months of follow-up after anti-TB treatment [18]. However, although this study lacked sufficient subjects to evaluate the effect of anti-TB treatment, the CCL11 level was not related to anti-TB treatment in the present study. On the other hand, CCL11 was not significantly different among HC, LTBI, and active TB (ATB) groups [19]. This study suggests the need to investigate the kinetics of CCL11 response in LTBI.

Next, we checked the mRNA expression levels of GM-CSF and TNFR. According to the study, GM-CSF is involved in supporting the productive phase of inflammatory responses [20,21]. The T-cell production of GM-CSF is required for the control of TB. It has been reported that the GM-CSF level was significantly different between MTB-infected and uninfected populations, and it has been suggested that GM-CSF could be the promising alternative biomarker to detect MTB [19]. In the present study, the mRNA expression levels of GM-CSF and TNFR were not significantly different among the HD/LTBI, HD/normal, and HC groups.

However, there were insufficient subjects in the 0.2 to 0.7 IU/mL response group and the existence of extreme values. These reasons could explain the different significance of mean and median values among the three QFT-GIT response groups in the HD and HC populations and the only HD population. Based on our results, mRNA expression levels of CCL11, GM-CSF, and TNFR require further investigation to distinguish LTBI and noninfected TB among immunocompromised patients in the future.

Although the reversion rate observed in the present study was not as high as the 72% observed in the United States (US), the reversion rate within six months observed in the present study was 34% regardless of the LTBI treatment [22]. In addition, the reversion rate of the 0.35 to 0.8 IU/mL response group was 74%, which was higher than that of the >0.81 IU/mL response group (24%) in the previous study [7]. Similarly, in the present study, the reversion rate of the 0.35–0.7 IU/mL response group was 85.7%, which was significantly higher than the >0.7 IU/mL response group (27.8%).

The reasons for IGRA variability are for the most part unexplained. Shu et al. (2013) suggested that the reversion occurs as an exponential decay, and the half-life of IGRA positivity was approximately two months and 18 months, respectively [23]. This phenomenon may be due to the attenuated cellular immunity in dialysis patients. However, in the present study, there was no significant difference in terms of demographic and clinical factors and mRNA expression levels of cytokines and chemokines between the reversion group and the persistent positive group. We could not match and analyze the differences of cytokines and chemokines using the previous samples of QFT-GIT tests within the past six months [24,25].

## 5. Conclusions

In conclusion, the present study revealed that the mRNA expression of CCL11 was statistically different in LTBI patients compared to HD patients without LTBI and HCs based on the current cut-off value (≥0.35 IU/mL). Additionally, there was a significant difference in the CCL11 mRNA expression level among the different QFT-GIT responses. Our study suggests that CCL11 may be a complementary or alternative biomarker for LTBI diagnosis, especially in immunocompromised patients.

The present study had certain limitations. The present study did not include an active TB group with HD and healthy donors. In addition, no peritoneal dialyzed patients with LTBI were included. The comparison of the mRNA expression level of cytokines and chemokines among the three different QFT-GIT test response groups was not efficient because of the insufficient number of subjects in the 0.2 to 0.7 IU/mL response group. Therefore, future large-scale studies and cohort designs are needed to evaluate the diagnostic performance of the cytokines and chemokines selected in the present study.

## Figures and Tables

**Figure 1 diagnostics-11-00595-f001:**
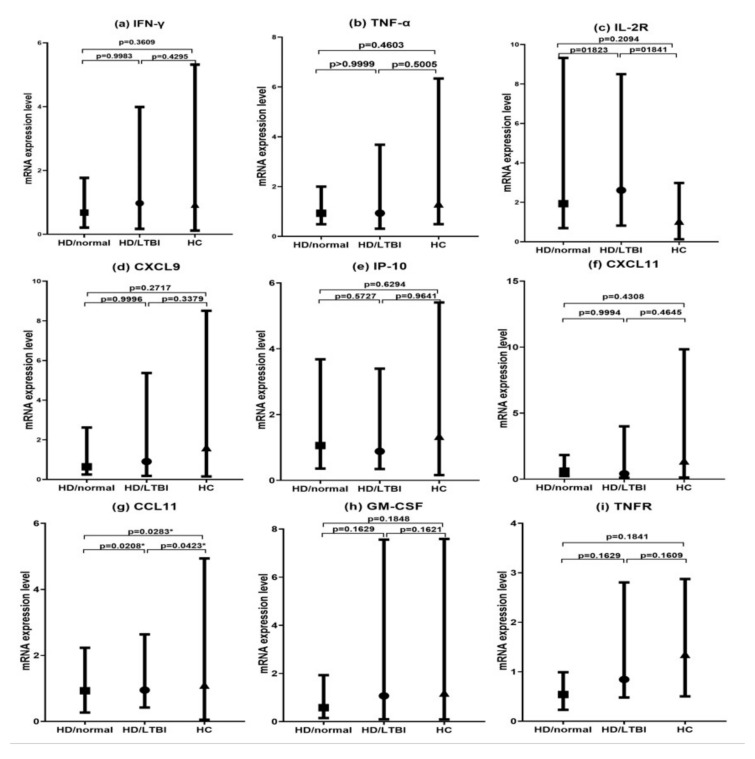
Target gene mRNA expression levels after stimulation of *Mycobacterium tuberculosis* (MTB)-specific antigens for 24 h according to latent tuberculosis infection (LTBI) diagnosis. Note: Whole blood of study subjects (hemodialyzed patients (HD) without LTBI (HD/normal) groups, HD with LTBI (HD/LTBI) group, and healthy control (HC) group) was stimulated with MTB-specific antigens (early secretory antigen target 6, culture filtrate protein 10, and TB 7.7) for 24 h and the mRNA expression levels for target genes were analyzed using real-time RT-PCR. The graphs represented median ± SD value (■, HD/normal; ●, HD/LTBI; ▲, HC) (**a**) interferon (IFN)-γ, (**b**) tumor necrosis factor-α (TNF-α), (**c**) IL-2R, (**d**) CXCL9, (**e**) IFN-γ-induced protein 10 (IP-10), (**f**) CXCL11, (**g**) CCL11, (**h**) granulocyte-macrophage-colony stimulating factor (GM-CSF), and (**i**) TNF receptor (TNFR). * *p* < 0.05.

**Figure 2 diagnostics-11-00595-f002:**
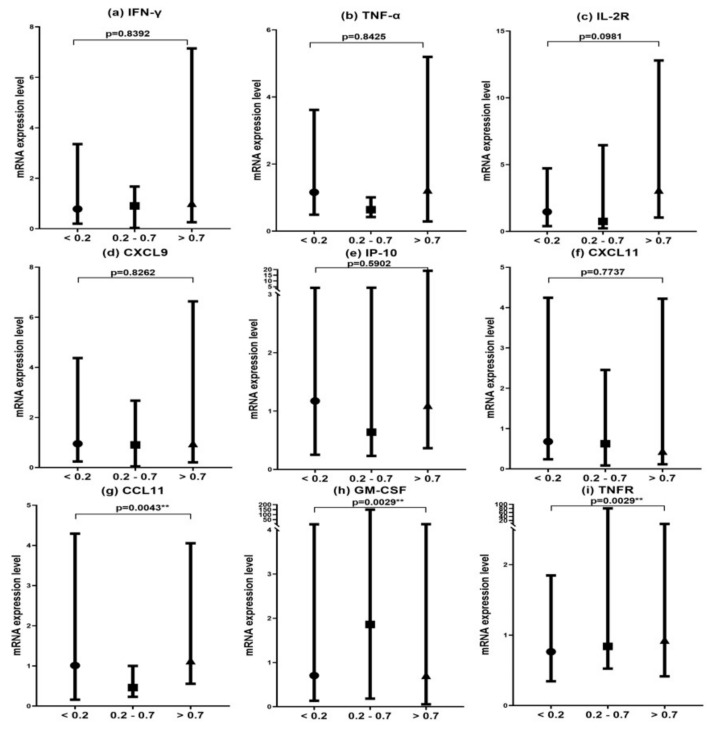
Target gene mRNA expression levels after stimulation of *Mycobacterium tuberculosis* (MTB)-specific antigens for 24 h according to QuantiFERON-TB Gold In-Tube (QFT-GIT) response groups. Note: Whole blood of study subjects (<0.2 IU/mL QFT-GIT response group, 0.2–0.7 IU/mL QFT-GIT response group, and >0.7 IU/mL QFT-IT response group) was stimulated with MTB-specific antigens (early secretory antigen target 6, culture filtrate protein 10, and TB 7.7) for 24 h and the mRNA expression levels for target genes were analyzed by real-time RT-PCR. The graphs represented median ± SD value (■, HD/normal;●, HD/LTBI;▲, HC) (**a**) interferon (IFN)-γ, (**b**) tumor necrosis factor-α (TNF-α), (**c**) IL-2R, (**d**) CXCL9, (**e**) IFN-γ-induced protein 10 (IP-10), (**f**) CXCL11, (**g**) CCL11, (**h**) granulocyte-macrophage-colony stimulating factor (GM-CSF), and (**i**) TNF receptor (TNFR). ** *p* < 0.01.

**Figure 3 diagnostics-11-00595-f003:**
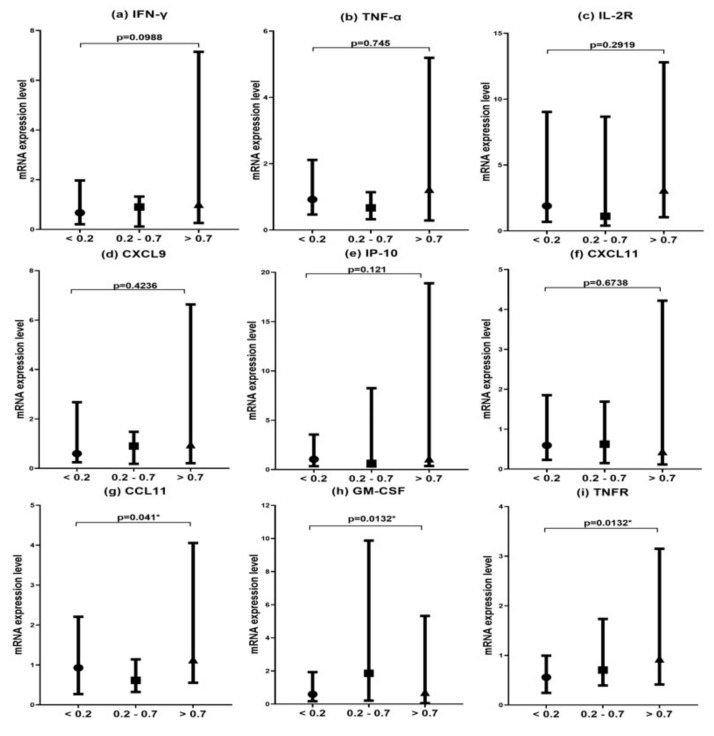
Target gene mRNA expression levels after stimulation of *Mycobacterium tuberculosis* (MTB)-specific antigens for 24 h according to QuantiFERON-TB Gold In-Tube (QFT-GIT) response groups in hemodialyzed patients (HD). Note: Whole blood of study subjects (the <0.2 IU/mL QFT-GIT response group, 0.2–0.7 IU/mL QFT-GIT response group, and >0.7 IU/mL QFT-IT response group) in HD was stimulated with MTB-specific antigens (early secretory antigen target 6, culture filtrate protein 10, and TB 7.7) for 24 h, and the mRNA expression levels for target genes were analyzed using real-time RT-PCR. The graphs represented median ± SD value (■, HD/normal;●, HD/LTBI;▲, HC). (**a**) interferon (IFN)-γ, (**b**) tumor necrosis factor-α (TNF-α), (**c**) IL-2R, (**d**) CXCL9, (**e**) IFN-γ-induced protein 10 (IP-10), (**f**) CXCL11, (**g**) CCL11, (**h**) granulocyte-macrophage-colony stimulating factor (GM-CSF), and (**i**) TNF receptor (TNFR). * *p* < 0.05.

**Table 1 diagnostics-11-00595-t001:** Demographics and clinical characteristics and QuantiFERON-TB Gold In-Tube (QFT-GIT) response results.

		Hemodialyzed (HD) Patients (*n* = 75)		Healthy Control ^c^ (*n* = 48)	*p*-Value
		LTBI ^a^ (*n* = 28)	Normal ^b^ (*n* = 47)		
Sex (Male)		18 (64.3%)	16 (34%)	21 (43.8%)	0.038 *
Age		62.00 ± 11.18	60.00 ± 12.46	24.00 ± 2.72	<0.001 ***
<65 years old		16 (57.1%)	30 (63.8%)	21 (100.0%)	0.796
65–74 years old		7 (25%)	11 (23.4%)	0 (0.0%)	
≥75 years old		5 (17.9%)	6 (12.8%)	0 (0.0%)	
Previous TB contact		7 (25%)	6 (12.8%)	0 (0.0%)	0.002 **
Previous TB treatment		0 (0.0%)	0 (0.0%)	0 (0.0%)	
BCG vaccination or scar		19 (67.9%)	34 (72.3%)	48 (100.0%)	<0.001 ***
Abnormal chest X-ray lesion		3 (10.7%)	0 (0.0%)	0 (0.0%)	0.005 **
Underlying diseases					
Diabetes mellitus		17 (60.7%)	20 (42.6%)	0 (0.0%)	0.100
Ischemic heart disease		9 (32.1%)	6 (12.8%)	0 (0.0%)	0.043 *
BMI (kg/m^2^)					0.026 *
<18.5		9 (32.1%)	4 (8.5%)	-	
18.5–22.9		9 (32.1%)	26 (55.3%)	-	
23.0–24.9		6 (21.4%)	5 (10.6%)	-	
25.0–29.9		4 (14.3%)	9 (19.1%)	-	
30.0–34.9		0 (0.0%)	3 (6.4%)	-	
Low BMI (<18.5 kg/m^2^)		9 (32.1%)	4 (8.5%)	-	0.012 *
Smoking		1 (3.6%)	5 (10.6%)	-	0.040 *
QFT-GIT results					
Positive		28 (100.0%)	0 (0.0%)	0 (0.0%)	
Indeterminate		0 (0.0%)	0 (0.0%)	0 (0.0%)	
Negative		0 (0.0%)	47 (100.0%)	48 (100.0%)	
Previous QFT-GIT results within 6 months					
Positive		27 (96.4%)	16 (34%)	-	<0.001 ***
Indeterminate		0 (0.0%)	0 (0.0%)	-	
Negative		1 (3.6%)	31 (66%)	-	
Reversion		0 (0.0%)	16 (37.2%)	-	
Conversion		1 (2.9%)	0 (0.0%)		
LTBI treatment history		6 (22.2%)	6 (37.5%)	-	0.232
QFT-GIT responses groups (IU/mL)					
Negative	<0.2	0 (0.0%)	46 (97.9%)	46 (95.8%)	<0.001 ***
Borderline	0.2–0.34	0 (0.0%)	1 (2.1%)	2 (4.2%)	
	0.35–0.7	7 (25%)	0 (0.0%)	0 (0.0%)	
Positive	>0.7	21 (75.0%)	0 (0.0%)	0 (0.0%)	
Previous QFT-GIT responses group within 6 months (IU/mL)					
Negative	< 0.2	1 (3.6%)	24 (51.1%)	-	<0.001 ***
Borderline	0.2–0.34	0 (0.0%)	7 (14.9%)	-	
	0.3–0.7	1 (3.6%)	6 (12.8%)	-	
Positive	> 0.7	26 (92.9%)	10 (21.3%)	-	
Plasma IFN-γ level after T cell mitogen stimulation (Mitogen-Nil, IU/mL)		9.40 ± 1.68	9.24 ± 1.34	10.00 ± 0.00	0.005 ** (a = b < c)

Note: Data were expressed as median ± SD for continuous variables and as percentages for categorical variables. *p*-values were calculated using one-way ANOVA tests with Dunnett’s multiple comparisons or Chi-square analysis. * *p* < 0.05, ** *p* < 0.01, *** *p* <0.001. Abbreviations: QFT-GIT, QuantiFERON-TB Gold In-Tube; TB, tuberculosis; BMI, body mass index; LTBI, latent tuberculosis infection; IFN, interferon. ^a^: HD/LTBI; ^b^: HD/normal; ^c^: healthy control. Statistical significance between hemodialysis patients and healthy control was confirmed.

**Table 2 diagnostics-11-00595-t002:** Comparison of QuantiFERON-TB Gold In-Tube (QFT-GIT) assay results between the current study and previous QFT-GIT within the past six months.

		QFT-GIT Responses (IU/mL)				
		<0.2	0.20–0.34	0.35–0.70	>0.7	Total
**Previous QFT-GIT responses within six months (IU/mL)**	<0.2	24 (96.0%)	0 (0.0%)	0 (0.0%)	1 (4.0%)	25 (100.0%)
	0.20–0.34	7 (100.0%)	0 (0.0%)	0 (0.0%)	0 (0.0%)	7 (100.0%)
	0.35–0.7	6 (85.7%)	0 (0.0%)	1 (14.3%)	0 (0.0%)	7 (100.0%)
	>0.7	9 (25.0%)	1 (2.8%)	6 (16.7%)	20 (55.5%)	36 (100.0%)
	Total (*n*)	46	1	7	21	75
Cohen κ coefficient = 0.386, *p* < 0.001						

Note: QFT-GIT, QuantiFERON-TB Gold In-Tube assay.

## Data Availability

The data analyzed during this study are included in this paper. Some of the datasets are available from the corresponding author upon reasonable request.

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
