# Peer review of "Cytokine and Chemokine mRNA Expressions after Mycobacterium tuberculosis-Specific Antigen Stimulation in Whole Blood from Hemodialysis Patients with Latent Tuberculosis Infection"

_diagnostics, 2021, doi:10.3390/diagnostics11040595_

Round 1

Reviewer 1 Report

Abstract: Results could be presented more clearly without statistical values such as "P = 0.028"

CCL11 abbreviation is not explain in the abstract, please add full title

Methods:

  • please correct SI units in all manuscript, especially "sec" to "s"
  • L119-128 - described details are needless, please shorten the sentences to "according to the manufactured protocols"
  • L135 - 140 - delete sentences, because described details are useless

Results:

  • L275-278 - please explain importance of this sentence or delete it

Discussion

  • please remove all mathematical or statistical values, because they have already been expressed in the results section
  • L342-344 - sentence can be deleted, because it is repeated

Author Response

Dear Editor, The Diagnostics

Thank you for your very good comments.

I made all responses to your comments and I changed figure and table forms in manuscript according to the reviewer’s comments and all changes were added with red color, and deleted sentences were highlighted with blue color.

Please check my revised article and responses.

Sincerely,

Sunghyun Kim

Reviewer 2 Report

This manuscript describes experiments aimed to analyze the mRNA expression levels of nine cytokines/chemokines in the whole blood samples after MTB-specific antigen stimulation from Hemodialysis patients and healthy control. The topic of the research is interesting, but the manuscript requires major revision.

Abstract

Lack of clearly defined aim of study.

Introduction

The introduction is written in a chaotic way and doesn’t introduce well enough into the topic. There is no clear link between sections.

Lines 37-38  What kind of  alterations in the immune system… the authors meant?

Lines 53-54 Could the authors precise the kinetics of what type of immune response they meant?

Materials and Methods

Point 2.1 line 75 Information on the number of blood samples should be included in this section.

Point 2.3 line 93 Instead of writing generally MTB-specific antigens a precise information including ESAT-6, CFP-10, TB 7.7 as well as PHA mitogen, should be indicated.

Lines 97-108 could be deleted and replaced with the appropriate citation as QFT-GIT assay is already well described in the literature

Point 2.6

The Authors stated in this section that ″geometric means and medians were used for measurements of central tendency″.  This statement is incorrect because the mean and median represent only the descriptive statistics. Other statistical parameters (for example the r-correlation coefficient) may define the ″tendency”. While the data in figures are expressed as median, the data in Table 1 are expressed as mean±SD. It would be better to standardize the type of descriptive statistics used in the manuscript.  

Results

Figures are presented as median value but the Authors did not indicate clearly whether this is median±SD or median±SEM??. A huge discrepancy in results  is visible, thus another graphic presentation of data should be used. What was the data distribution-normal or skew? The Authors should present the p value only to the data which are statistically significant.Point 3.5  The Authors mix mean value (line 272) and median value (line 274). This paragraph should be corrected to be less chaotic.

Discussion

Line 306-311 As the data were not statistically different, thus describing ˆtendency″ should be avoided.

Line 331 Please include the reference

In general a discussion is badly written, in most cases there are no links between sentences, the data interpretation in the context of literature is insufficient and unprecise.

The manuscript requires an English revision.

Author Response

(The authors gave the same response as above.)

Round 2

Reviewer 2 Report

The Authors have responded correctly to most of the comments. However, in result section still some not statistically significant p values are visible. I would suggest to remove it. Also, in lines 135-36 the sentence  'Geometric means and medians were used and data were expressed as mean ± SD for 135 continuous variables and percentages for categorical variables' is  confusing and should be changed. In Figures legends it is written that data are expressed as the median ± SD and interquartile range. This is truly unclear and should be clarified whether the bars show the SD or interquartile range.

Author Response

I deeply appreciate your time one more and careful review. In accordance with your comments, I have made the appropriate revisions to the manuscript.

At first, I checked my manuscript again, however, we could not find statistically insignificant p value in the manuscript. Therefore, if you let us know the which line of page include statistically insignificant p value, we will remove it.

Next, I changed sentence that could be confusing with other sentences. When writing the data, I tried to explain that the figure bars were written interquartile range of the total data, but I thought it could confuse readers when reading this paper. So, I changed sentence.

“The data was expressed as median± SD and for the further confirmation of statistical reliability, interquartile range of the total data was used.

Last, in figure, I deleted unclear sentence in all figure “with interquartile range (bar)”.